# Discrete choice experiment exploring women's preferences in a novel device designed to monitor the womb environment and improve our understanding of reproductive disorders

Ka Ying Bonnie Ng  ,[1,2] Rhiannon Evans,[3] Hywel Morgan,[4] Emmanouil Mentzakis,[5] Ying C Cheong[1,2]

For numbered affiliations see end of article.

**Correspondence to**
Dr Ka Ying Bonnie Ng;
bonnie.ng@doctors.org.uk

## ABSTRACT

**Objectives** The study aims to determine the relative importance of key attributes of a novel intrauterine device. The device monitors uterine oxygen, pH and temperature in real time with the aim of improving our understanding and treatment of reproductive disorders.

**Design** A discrete choice experiment was used to elicit preferences in this novel investigative tool. The attributes and levels used in the choice scenarios were length of time using the device (7, 14 or 28 days), information obtained to guide treatment (limited, majority or all cases), risk of complications (1% or 10%) and discreteness (completely discrete, moderately discrete or indiscrete).

**Setting** Secondary care hospital in Hampshire, UK.

**Participants** 361 women of reproductive age.

**Primary and secondary outcome measures** Conditional logit and latent class logit regression models to determine the preference for each attribute.

**Results** Conditional logit coefficients allow comparison between attributes; women placed most importance on obtaining information to guide treatment in all cases (2.771), followed by having a completely discrete device (1.104), reducing risk of complications by 1% (0.184) and decreased length of time by 1 day (0.0150). All coefficients p<0.01. Latent class conditional logit assigns participants to two classes with 27.4% in class 1 who are less likely to have higher education or qualify for National Health Service-funded in vitro fertilisation compared with class 2. Those in class 2 placed 1.7 times more importance on a device whose information guided treatment in all cases and a 1% decrease in complications risk was nearly 15 times more attractive.

**Conclusions** Women placed most importance on having a device that obtains information to guide treatment and are willing to use the device for a longer, have a device with higher risk of complications and an indiscrete device if it is able to provide answers and direction for treatment of their reproductive disorder.

## INTRODUCTION

The development and introduction of novel technologies, diagnostic tools and therapies in the area of clinical medicine is rapidly

## Strengths and limitations of this study

► This discrete choice experiment (DCE) explores women's preferences for a novel intrauterine-monitoring device designed as an investigative tool in women with reproductive failures.

► DCEs allow for the elicitation of preferences over characteristics and their trade-offs for (not yet in the market) devices in a structured and robust manner.

► Participants were recruited in a single UK hospital, which may limit the applicability in other parts of the UK or in other countries.

► Inherent to DCE, participants do not experience the resulting consequences of their decisions, therefore their responses may not accurately reflect real choices.

► We have not included a cost attribute as our aim was to explore the trade-offs between the attributes, which seemed most important to users rather than economic analysis.

increasing. Within reproductive health, we have seen the recent introduction of technological tools such as Ovusense, skin or vaginal sensors designed to monitor menstrual cycles and track ovulation while a woman is trying to conceive and AneVivo, a novel device that enables in vitro fertilisation (IVF) to take place in the maternal womb instead of a laboratory incubator for women who are having assisted reproductive technologies. Despite the rapid development of novel diagnostic and therapeutic tools, including medical devices to assist with conception and pregnancy, there are limited data exploring user's preferences. Engagement of users in the development of novel devices is lacking yet hugely important to inform scientists, researchers, clinicians and policy-makers. Most importantly, engagement of users in the development of novel

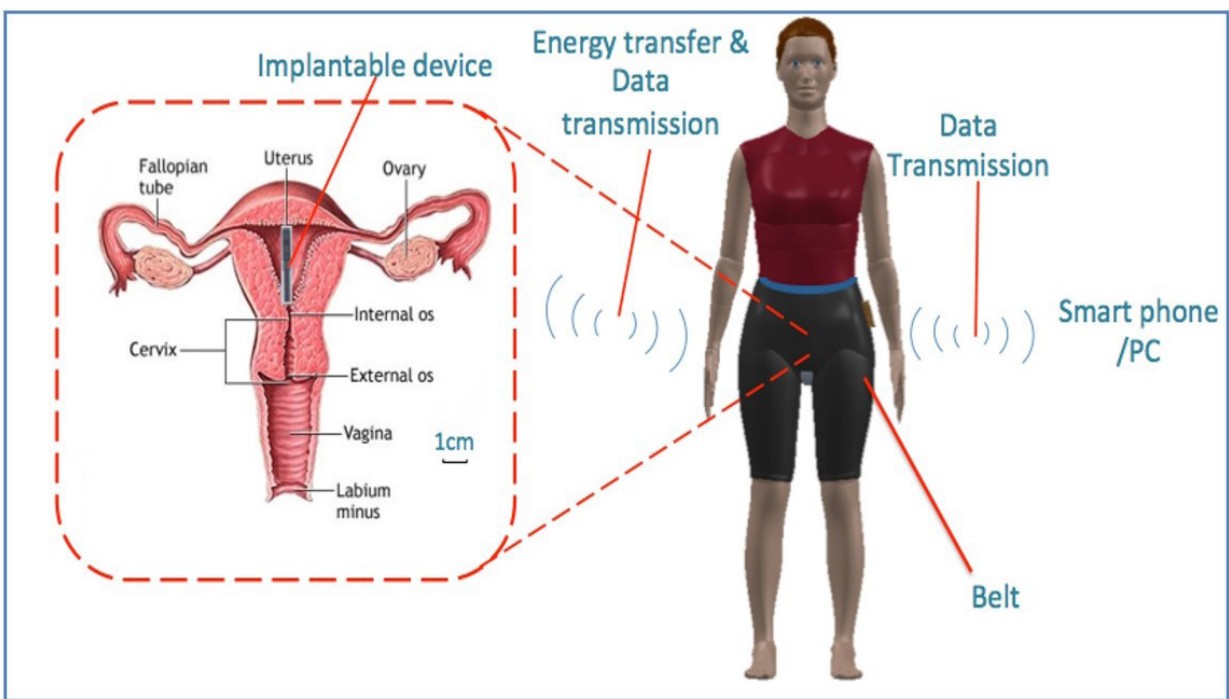

**Figure 1** Pictoral representation of the novel intrauterine device and on how it will sit within the uterus. The sensor is batteryless and is powered using a belt and a mobile phone. A belt worn around the waist delivers energy to the sensor and collects data when required. A smartphone/PC controls the system and collects the data.

therapies and devices would inform and benefit the users themselves.

Within health economics, discrete choice experiment (DCE) is a quantitative technique used for eliciting individual preferences allowing researchers to explore how individual's value attributes of a programme, product or service. DCEs proceed by asking individuals to state their choice over different hypothetical alternatives over a series of scenarios. It is used increasingly in health services research, primarily to assess patient-stated preferences and willingness to pay for different models of health service delivery.[1–3] Within the area of reproductive medicine, DCEs have been used to elicit the preferences placed on fertility treatment,[4] including the value of patient centred care[5 6] and willingness to accept risk and complications of fertility treatment.[7]

Recently, an implantable batteryless sensor, which can be placed within the uterus to measure oxygen, pH and temperature, has been developed in collaboration with a technology company (Verso Biosense, patent number PCT/GB2017/050609) and University of Southampton (figure 1).

The device is similar in shape and size to a contraceptive coil and has an accompanying wearable garment with an information receiver to collect data. This device has the potential to help personalise and streamline care for women with reproductive disorders such as infertility or repeated miscarriages. Current treatment for women with reproductive disorders remains suboptimal and there is no ability to triage patients into appropriate disease/

treatment groups. Up to 20% of couples with infertility and 50% of women with recurrent miscarriages have no identified cause for the reproductive problem and as such are then offered either blanket IVF treatment or supportive care. Additionally, IVF treatment is not always successful; the average birth rate per embryo transferred is 24% in 2018.[8] Endometrial receptivity and the uterine microenvironment are key areas, which warrant further investigation to advance the care for women in the field of reproductive medicine.

A number of key attributes for the reproductive device including availability of the device, safety, effectiveness as an investigative tool and inconvenience were previously identified from early patient and public involvement (PPI) work. However, there remains a need to understand what value potential users place on such key attributes, and to explore the relative importance of each attribute if this novel intrauterine device is to be offered as an investigative tool for women affected by reproductive failures in the future.

The aim of this study is to elicit patient preferences for the attributes/characteristics of a novel intrauterine-monitoring device as an investigative tool in women with reproductive failures by using a DCE. Insight into the value that women place on key attributes can assist with the further design or development of the device, and the design of clinical trials using the device. The findings may also guide researchers and clinicians designing novel devices within the area of reproductive medicine.

**Table 1** Attributes and levels for the novel intrauterine-monitoring device

| Attributes | Description | Levels |
|---|---|---|
| Length of time using the device | The number of days the device would remain inside your womb. Longer time in your womb could give more information to the doctor. There are three choices | 7 days<br>14 days<br>28 days |
| Information obtained from the device and its use in guiding treatment | There are three choices, the information obtained will guide treatment in a limited number of cases, majority of cases or all cases | Limited number of cases=the device would be able to gather information to guide treatment for some women with a reproductive disorder<br><br>Majority of cases=the device would be able to gather information to guide treatment for most women with a reproductive disorder<br><br>All cases=the device would be able to gather information to guide treatment for anyone who uses it regardless of their reproductive condition |
| Risk of complications | There may either be short-term or long-term complications from using the device. The long-term effects of using the device are currently unknown, and further clinical trials will be conducted to explore this. The short-term complications could be an infection needing antibiotic treatment, pain or discomfort needing pain relief or expulsion (device falling out of the womb). There will be two choices | 1% lower risk<br>10% higher risk |
| Discreteness of the information receiver | Once the sensor is inserted into the womb, you will be asked to wear a garment which holds the information receiver. There are three options | Completely discrete=even when tight clothing is worn the receiver will not be noticeable to others<br><br>Moderately discrete=the receiver will be noticeable to others when tight clothing is worn<br><br>Indiscrete=the receiver will be noticeable to others irrespective of the type of clothing worn |

## METHODS

### Study design

This study used a DCE to elicit patient preferences regarding a novel intrauterine investigative tool to help them with the treatment of their infertility/recurrent miscarriages.

### Development of attributes and levels

Each hypothetical sensor was being described to the participant in terms of four attributes ((1) length of time using the device, (2) information obtained from the device and its use in guiding treatment, (3) risk of complications and (4) discreteness of the information receiver) summarised in table 1. These were derived from a PPI session, a literature search and discussion with experts (three clinicians and researchers working within reproductive medicine and one health economist) as to which attributes would be of importance. The attributes and DCE Questionnaire were further refined through preliminary testing of the questionnaire with potential users of the device.

### DCE design

The combination of attributes and levels resulted in 54 (3×3×2×3) possible scenarios. However, given cognitive burden and fatigue for the respondents, the number of questions to be asked had to be reduced and hence we opted for a fractional factorial design. For this study, we were interested in estimating effects for four attributes as

well as all two-way interactions with the information attribute. As such, the number of estimable parameters to be identified is 18 (ie, 17 plus 1 for the constant). A D-efficient fractional factorial design with 18 pair-wise choice sets was created in SAS software (Version 9.4). The literature suggests that 18 choices per individual is possibly more than respondents can handle, potentially impacting the quality of responses.[5] To reduce cognitive burden, the design was blocked into two blocks of nine choice sets each. At the end of the experiment, a series of questions on patients' demographic and medical characteristics was included.

The preliminary DCE Questionnaire was piloted on 10 women within the obstetrics and gynaecology department. The women were interviewed: the questionnaire's comprehensibility, usability, amount of choice sets and content validity of the attributes and levels were discussed. After the pilot testing, changes were implemented and repeat testing was carried out. Feedback from the pilot testing shaped the exact formulation of the attributes and the DCE instructions on the final DCE Questionnaire (online supplemental appendix 1).

### Recruitment of participants

Participants were women of reproductive age (18–50 years), who were wishing to achieve a pregnancy now or sometime in the future. We have included all women and

have not limited inclusion of women with reproductive disorders as in future clinical studies, we plan to trial the device in women without reproductive disorders as well as those with a range of reproductive problems in order to determine the uterine environment in physiological as well as pathological conditions. Exclusion criteria included those with limited English or limited ability to understand the study design and questionnaire (eg, language barrier, learning disability, inability to process the information to complete the questionnaire). Participants who met the inclusion and exclusion criteria were recruited from gynaecology clinics within Princess Anne Hospital, Southampton. Women who agree to participate in the study were asked to complete the DCE Questionnaire pack (hard copy).

Precise power calculation for DCEs have not been developed, but some rule-of-thumb approaches are available.[9] The first suggest that there are at least 50 respondents per alternative, that is, 100 for our design. The second posits that sample size should be larger than $500 \times c/(t \times a)$, where c is the largest number of levels of attribute (or levels of interaction), 't' is the number of choice tasks and 'a' is the number of alternatives. Hence, for main effects, only estimation we would need around 42 individuals per block (ie, 84 in total), while for main plus our specified two-way interactions we would need around 125 individuals per block (ie, 250 in total).

## Data collection

The choice sets were administered in an anonymous hard copy of the questionnaire pack (online supplemental appendix 1). The questionnaire pack consisted of several sections including participant information about the study, consenting to the questionnaire, explanation about the DCE Questionnaire and how to complete it, an example question, choice sets, 'about yourself' and an optional feedback section. Within the 'explanation about the questionnaire' section, the attributes and levels were explained. Each choice set consisted of two novel intrauterine device alternatives described according to the four attributes in the format as in table 2. The participants

**Table 2** An example choice set presented to participants

|  | Device A | Device B |
|---|---|---|
| Length of time using the device | 7 days | 28 days |
| Information guides treatment | All women | All women |
| Risk of complications | 10% | 1% |
| Discreteness | Indiscrete | Moderately discrete |

Within each choice set, participants are asked to consider the devices described by the two sets of attributes and choose between device A or device B. Each hypothetical device presented will vary in the four attributes: length of time using the device, information obtained from the device to guide treatment, risk of complications and discreteness.

completed the questionnaire themselves and were given oral clarification where required. Each participant was assigned a study ID.

Participants were asked to select their most preferred option. Finally, there was an optional section at the end of the questionnaire asking participants about their sociodemographic characteristics, with a range of questions informed by those characteristics, which have been found to affect fertility/pregnancy or were hypothesised to influence treatment uptake.

To set up the choice context, participants were asked to imagine that their doctor had given them the option of using a novel intrauterine sensor device to find out more about their womb environment, which may help guide them in making decisions about next treatment steps towards a successful pregnancy. They were asked to choose the preferred device in each choice set if these were offered as investigation options in 'real life'.

## Analysis of data

We performed econometric analysis from the responses obtained from the DCE Questionnaire. Conditional logit models were initially estimated. Unobserved preference heterogeneity was explored through a latent class model (LCM) whereby individual are probabilistically sorted into classes whose number is exogenously set. LCM allows identification of distinct behavioural and preference patterns within the data and different regression parameters are estimated for each class.[9 10] Further, we parameterise the class membership equation to assess whether given individual characteristics explain which individuals fall within which class.

Coefficients in DCE models capture whether an attribute level increases or decreases the attractiveness (or utility in economic terms) of a device featuring such characteristics. However, coefficients (due to variance normalisation) lack direct quantitative interpretation. Yet, within each model and across classes, the relative importance of coefficients is feasible, while interpretation can also rely on ratios of coefficients.

To provide further, intuitive interpretation of results, we calculate and present predicted probabilities for two sets of eight hypothetical devices with varying characteristics, ranging from the one that would seem to be the most attractive to the least attractive. The resulting probabilities essentially give the probability that any given device would be selected by respondents if the eight presented were the available options.

## Patient public involvement

Engagement of patients is critical to the design of the DCE Questionnaire, and this was introduced from the conception of the study. The key attributes and levels were identified through a series of focus groups involving potential users for the novel intrauterine device. In the focus groups, women with reproductive failures (subfertility or repeated miscarriages) were asked for their views

of the device and their potential involvement in a clinical trial using the device.

A preliminary DCE Questionnaire was piloted on 10 women within the obstetrics and gynaecology department. Further interviews with the women refined the questionnaire's comprehensibility, usability, amount of choice sets and content validity of the attributes. Feedback from the PPI work shaped the exact formulation of the attributes and the DCE instructions on the final DCE Questionnaire.

## RESULTS

### Study participants

Overall, 361 women were recruited into this DCE Questionnaire Study. Table 3 summarises the participant characteristics. The mean age of the participants was 32.5 years (SD 5.23). Most of the participants (60.1%) were married, 23.9% were in a relationship, 11% were engaged and 3.8% were single. Over half of the participants (52.3%) had higher education and professional/vocational equivalents. Overall, 90.6% of participants were British, 4.7% were Asian or Asian British, 1.7% were of mixed ethnicity and 0.8% were black/African/Caribbean or black British. 81.9% of the women were needing treatment for infertility and 5.4% were needing treatment for recurrent miscarriage. 22.3% of participants or their partner had already had a child. Overall, 30.4% qualified for National Health Service (NHS)-funded IVF.

### Conditional logit

The results of the conditional logit are presented in table 4. Two-way interactions with the information attribute were not statistically significant and did not reveal significantly different preference patterns. Further, we test the need for alternative-specific constant (ie, whether respondents for some reasons preferred devices presented first over those presented second, irrespective of characteristics) and confirm its lack of statistical significance. Hence, we focus on main effects models without alternative-specific constants. Finally, using a heteroskedastic logit specification, we test for scaling differences between the blocks and again confirm lack of evidence for any differences allowing us to pool and analyse the data.

From the conditional logit coefficients, it is possible to make relative comparisons in the magnitude of the coefficients, comparing between the effect of different attributes and attribute levels. Women placed the most importance on the attribute of obtaining information to guide treatment, followed by discreteness of the device and its receiver, risk of complications and lastly length of time of use.

Having a device that provides information that guides treatment in all cases (coefficient 2.771) is over two times more attractive as a device, which provides information in majority of cases (coefficient 1.243). Women placed the same level of importance in a device that provides information that guides treatment in all cases

(coefficient 2.771) and a device whereby the risk of complications (coefficient 0.184 per 1% decrease) is reduced by 15%. Completely discrete devices (coefficient 1.104) were over 1.5 times more attractive than moderately discrete devices (coefficient 0.793). Women

**Table 3** Summary of study participants

| | | Mean | SD |
|---|---|---|---|
| Age | | 32.5 | 5.23 |
| | | **N** | **%** |
| Relationship status | Single | 7 | 3.8 |
| | In a relationship, living apart | 3 | 1.5 |
| | In a relationship, living together | 41 | 22.4 |
| | Engaged | 20 | 11.0 |
| | Married | 110 | 60.1 |
| | Civil partnership | 1 | 0.6 |
| | Living with parents | 1 | 0.6 |
| Educational attainment | Higher education and professional/vocational equivalents | 184 | 52.3 |
| | A levels, vocational level 3 and equivalents | 81 | 23.0 |
| | GCSE/O level grade A×–C, vocational level 2 and equivalents | 62 | 17.6 |
| | Qualifications at level 1 and below | 9 | 2.6 |
| | Other qualifications: level unknown (including foreign qualifications) | 9 | 2.6 |
| | No qualifications | 7 | 2.0 |
| Ethnicity | British | 327 | 90.6 |
| | Asian/Asian British | 17 | 4.7 |
| | Mixed ethnicity | 6 | 1.7 |
| | Black/African/Caribbean /black British | 3 | 0.8 |
| | Other | 3 | 0.8 |
| Needing treatment for | Infertility/subfertility | 289 | 81.9 |
| | Recurrent miscarriages (three or more consecutive early miscarriages) | 19 | 5.4 |
| Either the female participant or her partner have a child | | 79 | 22.3 |
| Qualify for NHS-funded in vitro fertilisation | Yes | 105 | 30.4 |
| | No | 92 | 26.6 |
| | Unsure | 149 | 43.1 |

GCSE, General Certificate of Secondary Education; GCSE/O, General Certificate of Education Ordinary Level 'O level'; NHS, National Health Service.

put the same level of importance on 12 days decrease in the length of use of the device (coefficient 0.015 per 1 day decrease) as a 1% decrease in the risk of complications from using the device (coefficient 0.184 per 1% decrease).

## Latent class conditional logit

The latent class conditional logit analysis probabilistically assigns participants to two classes with 27.4% of participants falling within class 1% and 72.6% within class 2 (table 4). In general, class 1 participants are less likely to have higher education (coefficient −0.660, SE 0.307, 95% CI −0.967 to −0.353), and less likely to qualify for NHS-funded IVF (coefficient −0.709, SE 0.356, 95% CI −1.065 to −0.353).

### Class 1

Having a device that provides information that guides treatment in all cases (coefficient 1.996) is almost two times as attractive as a device, which provides information in majority of cases (coefficient 1.021). Moving from a completely discrete device (coefficient 1.250) to a moderately discrete device (coefficient 0.807) reduced the attractiveness by a third. Length of use of the device and risk of complications are not significant factors altering participant's preference for the device.

### Class 2

Having a device that provides information that guides treatment in all cases (coefficient 3.456) is 1.7 times more attractive than a device, which provides information in majority of cases (coefficient 2.008). Women put the same level of importance on 12 days decrease in the length of use of the device (coefficient 0.0271 per 1 day decrease) as a 1% decrease in the risk of complications from using the device (coefficient 0.318 per 1% decrease). Obtaining information that guided treatment in all cases (coefficient 3.456) had an equal importance to an 11% decrease in the risk of complications (coefficient 0.318 per 1% decrease).

Obtaining information that guided treatment in all cases (coefficient 3.456) had an equal importance to a reduction in the length of use (coefficient 0.0271 per 1 day decrease) by 128 days. There was a slight decrease in the attractiveness of the device when the receiver was moderately discrete (coefficient 1.349) compared with completely discrete (coefficient 1.462).

### Comparing class 1 to class 2

Those in class 2 placed more importance on a device that provided information to guide treatment. Compared with the baseline, obtaining information to guide treatment in all cases was 1.7 times more important in class 2 (coefficient 3.456) compared with class 1 (coefficient 1.996).

**Table 4** Women's preferences in the four features of the intrauterine-monitoring device: length of time using the device (per day decrease), information obtained from the device to guide treatment, risk of complications (per 1% decrease) and discreteness

| Preferences | Conditional logit | Latent class conditional logit | |
| --- | --- | --- | --- |
| | | Class 1 | Class 2 |
| Length of time (1 day decrease) | 0.0150*** (0.00425) | 0.0111 (0.00682) | 0.0271*** (0.00935) |
| Information (baseline: limited) | | | |
| Majority | 1.243*** (0.142) | 1.021*** (0.333) | 2.008*** (0.228) |
| All | 2.771*** (0.209) | 1.996*** (0.349) | 3.456*** (0.287) |
| Risk of complications (1% decrease) | 0.184*** (0.00978) | 0.0216 (0.0161) | 0.318*** (0.0206) |
| Discreteness (baseline: indiscrete) | | | |
| Completely | 1.104*** (0.0991) | 1.250*** (0.158) | 1.462*** (0.190) |
| Moderately | 0.793*** (0.0905) | 0.807*** (0.164) | 1.349*** (0.222) |
| Class probabilities | | | |
| Higher education | | −0.660** (0.307) | – |
| Qualify for National Health Service-funded in vitro fertilisation | | −0.709** (0.356) | – |
| Constant | | −0.463* (0.252) | – |
| Class shares | | 0.274 | 0.726 |
| Obs, n | 6496 | 6190 | |
| Resp, n | 361 | 344 | |
| BIC | 2639 | 2425 | |

Coefficients for the conditional logit and latent class conditional logit are shown along with robust standard errors in parentheses; ***p<0.01, **p<0.05, *p<0.1.
BIC, Bayesian Information Criteria.

**Table 5** Features and predicted probabilities of eight hypothetical devices (A–H) being chosen according to conditional logit

| Hypothetical device | A | B | C | D | E | F | G | H |
|---|---|---|---|---|---|---|---|---|
| **(A)** | | | | | | | | |
| Length of time using the device | 7 days | 7 days | 14 days | 7 days | 7 days | 14 days | 7 days | 28 days |
| Information obtained from device and its use in guiding treatment | All cases | Majority of cases | Majority of cases | All cases | Limited cases | Limited cases | Limited cases | Limited cases |
| Risk of complications | 1% | 1% | 1% | 10% | 1% | 1% | 10% | 10% |
| Discreteness of the information receiver | Completely | Completely | Moderately | Moderately | Completely | Moderately | Moderately | Indiscrete |
| **Predicted probabilities (conditional logit)** | **61.90%** | **13.40%** | **8.90%** | **8.60%** | **3.90%** | **2.60%** | **0.50%** | **0.20%** |
| **(B)** | | | | | | | | |
| Length of time using the device | 7 days | 14 days | 7 days | 14 days | 7 days | 14 days | 7 days | 14 days |
| Information obtained from device and its use in guiding treatment | Limited cases | Limited cases | Limited cases | Limited cases | Limited cases | Limited cases | Limited cases | Limited cases |
| Risk of complications | 1% | 1% | 1% | 1% | 10% | 10% | 10% | 10% |
| Discreteness of the information receiver | Completely | Completely | Moderately | Moderately | Completely | Completely | Moderately | Moderately |
| **Predicted probabilities (conditional logit)** | **25.50%** | **23.00%** | **18.70%** | **16.80%** | **4.90%** | **4.40%** | **3.60%** | **3.20%** |

(A) shows various combinations of attribute levels, ranging from the most preferred device to the least preferred device attribute levels.
(B) shows various combinations of attribute levels of the device, fixing one attribute, information obtained device and its use in guiding treatment to a 'limited' number of cases.

Obtaining information to guide treatment in majority of cases was two times more important in class 2 (coefficient 2.008) compared with class 1 (coefficient 1.021).

Class 2 participants placed more importance on length of use for the device. A device that has an increased length of use (by 1 day) is nearly 2.5 times less attractive for class 2 (coefficient 0.0271 per 1 day decrease) than class 1 (coefficient 0.0111 per 1 day decrease). Class 2 participants placed more importance on the risk of complications. A device that had a 1% decrease in the risk of complications is nearly 15 times more attractive for class 2 (coefficient 0.318 per 1% decrease) than class 1 (coefficient 0.0216 per 1% decrease). Class 2 participants placed a slightly higher importance on having a discrete device. Having a completely discrete device was more attractive for class 2 (coefficient 1.462) than class 1 (coefficient 1.250).

### Predicted probabilities

Given that the results in both the conditional logit and the latent class conditional logit have a similar pattern of preferences, with some differences in the magnitudes between the two classes, we focus on the predicted probabilities of the conditional logit. Predicted probabilities indicate the probability that any one of the eight hypothetical devices described will be chosen when all eight are on offer (table 5). The importance of the combinations of the characteristics of each of these hypothetical devices is represented by the magnitude of their probabilities.

When eight hypothetical devices are presented to an individual with varying levels of different attributes (table 5A), over 60% of the time, hypothetical device A will be chosen (shortest length of time using the device (7 days), obtaining information that guided treatment in all cases, lowest risk of complications (1%) and completely discrete). About 13% of the time, they are predicted to choose hypothetical device B (7 days, obtaining information that guided treatment in majority of the cases, 1% risk and completely discrete). About 9% of the time, an individual would choose device D (7 days, obtaining information that guided treatment in all cases, 10% risk and moderately discrete).

Next, we assess the effect on predicted probabilities when the devices presented were all able to provide information to guide treatment in a limited number of cases, with varying levels of the other attributes. When eight hypothetical devices with the information attribute

fixed to guiding treatment in a limited number of cases are presented (table 5B), about 25% of the time, an individual would choose device A (7 days, 1% risk, completely discrete), 23% of the time, they would choose device B (14 days, 1% risk, completely discrete) and 19% of the time, they would choose device C (7 days, 1% risk, moderately discrete). The four devices with the higher risk of complications (10%) had a low likelihood (<5%) of being selected.

## DISCUSSION

To our knowledge, this is the first DCE study of women's preferences for a novel intrauterine-monitoring device as an investigative tool designed to monitor the womb environment. We have shown that all four selected attributes played a significant role in women's preferences for the intrauterine-monitoring device. The ideal attributes of the device would be that it is used for the shortest length of time, guide treatment in all cases, have the lowest risk of complications, and be completely discrete. In terms of importance of the device attributes, the ability of the device to obtain information and guide treatment was the most important, followed by discreteness of the device, then risk of complications and finally length of using the device. Our study provides insights into the preferred attributes of a novel device, and the influence of sociodemographic characteristics on choices women make. These findings will help with the further development of the monitoring device and to ensure that the women's preferences are considered prior to the design of clinical trials using the device.

Women were willing to have a device for a longer length of time, with higher risk of complications or an indiscrete device if it was able to provide information that guides treatment in all cases. The importance of the device being able to guide treatment is emphasised by the fact that its attractiveness is doubled when the device can guide treatment in all cases, rather than only in a majority of cases. Currently, about 15%–20% of couples who are being investigated for infertility and about 50% of couples being investigated for recurrent miscarriage have no known cause for their reproductive problems and are deemed as 'unexplained' after their diagnostic workup.[11 12] This tends to lead to frustration for the couples, and it seems that a device which can guide treatment in all cases of reproductive failures is most preferred. At present, unexplained infertility is managed by offering blanket IVF treatment to couples as current investigative tools are not able to triage the patients into the appropriate disease/treatment groups. In the cohort of patients with unexplained recurrent pregnancy loss, the main emphasis following on from a negative diagnostic workup is supportive care.[13] Our inability to find a cause of the couple's infertility or repeated miscarriages does not mean that there is no cause for the disorder; endometrial receptivity and the uterine environment are key areas that warrant further investigation in search for possible treatable causes and personalised treatment.[14] A reproductive investigative tool is much more likely to be opted for if it helps to provide answers to help guide treatment.

Discreteness of a fertility assessment tool plays an important role in women's preferences; having a device that is completely discrete is over 1.5 times more attractive than a device that is moderately discrete. This effect is more pronounced in the cohort of women who fall in a category where they are more likely to have higher education and more likely to qualify for NHS-funded IVF. Women who have reproductive disorders often feel isolated and stigmatised[15 16] and thus the value they place on 'not being visible' in society is high. When women are affected by infertility, many choose not to share their problem with others including family members because of the stigmatising and repressive perspectives of society towards people who can not conceive naturally and towards IVF treatment.[17] Moreover, despite psychological counselling being offered to patients with reproductive disorders, the uptake is low.[18] These features tend to support our findings of the infertile women not wanting to be 'highlighted' or 'labelled' by an indiscrete reproductive investigative device in society.

Women valued the importance of having a device that has the lowest risk of complications; they were willing to use the device for almost 2 weeks longer if it was able to lead to a 1% reduction in the risk of complications from the device. Given risk perceptions involve incorporating numerical information about threat, the ability to understand and use the numeric information presented plays a significant role in the formation and use of risk perceptions.[19] Individuals who are well educated are more likely to retrieve and use numerical principles in decision-making, and within our analyses, we have shown that women who are more likely to fall in the category who have had higher education place more importance on a device with lower risk of complications (1% reduction in the risk of complications is 15 times more important to the group of women who are more likely to fall within the group who have had higher education).

The length of time using the device does not seem to play a major role in women's preferences towards the use of the novel investigative tool for reproductive disorders. Within our cohort of women, if the device is able to provide information to guide treatment in all cases, they were willing to use the device for up to 184 days.

## Limitations

Participants were recruited in a single UK hospital, which may limit its applicability in other parts of the UK or in other countries. Future research could compare women's preferences across different populations, cultures and health service provisions (eg, NHS or private patient cohorts).

Although DCEs have been widely used in many areas of healthcare to elicit patient preferences, they have important limitations. Participants are evaluating the hypothetical choice sets intended to simulate real-life decisions, but the responses may not accurately reflect real choices as participants do not experience the resulting consequences of their decisions. However, the fact that results are in line with prior expectations mitigates such concerns. Another limitation is the restricted number of attributes and levels that can be included in the choice sets to reduce cognitive burden and for it to be practically feasible. We have also not included a cost attribute as our aim was to explore the trade-offs between the attributes which seemed most important to users from PPI work that was performed prior to the study, rather than include economic analysis. To quantify trade-off in one attribute relative to another, we have assumed that the levels are linear, for example, the risk of complications is linear between the two levels (1% and 10%); in reality the risk preferences between these levels may not be linear.

## CONCLUSIONS

Within women planning a pregnancy, all four selected attributes played a significant role in altering women's preference for a novel intrauterine device designed to monitor the womb environment. Women seemed to place most importance on having a device that obtains information to guide treatment and are willing to use the device for a longer length of time, have a device with higher risk of complications and an indiscrete device if it is able to provide answers and direction for treatment of their reproductive disorder. Women who are more likely to have higher education and qualify for NHS-funded IVF place more importance on all attributes.

**Author affiliations**
¹School of Human Development and Health, University of Southampton Faculty of Medicine, Southampton, UK
²Department of Obstetrics and Gynaecology, Princess Anne Hospital, Southampton, UK
³Department of Obstetrics and Gynaecology, Salisbury NHS Foundation Trust, Salisbury, UK
⁴Department of Research, Electronics and Computer Science, University of Southampton, Southampton, UK
⁵Department of Economics, University of Southampton, Southampton, UK

**Acknowledgements** The authors would like to thank the obstetrics and gynaecology research team (with special mention to Susan Wellstead and Agnieszka Burtt) at the Princess Anne Hospital, and the gynaecology outpatient team at the Complete Fertility Unit for their hard work and help with conducting the study. We would also like to extend our gratitude to all the women who took part in the study.

**Contributors** KYBN and YCC conceived the idea. KYBN, HM, EM and YCC deisgned the experiments KYBN and RE recruited participants into the study. All coauthors analysed and interpreted the data. KYBN and RE wrote the initial draft of the paper and the final version was approved by all coauthors prior to submission. YCC is the guarantor.

**Funding** This study has been funded by the NIHR i4i Innovation grant (grant number II-LB-0715-20002) awarded to YCC.

**Competing interests** YCC is a co-founder, non-executive director and minority shareholder of Verso Biosense.

**Patient and public involvement** Patients and/or the public were involved in the design, or conduct, or reporting, or dissemination plans of this research. Refer to the Methods section for further details.

**Patient consent for publication** Not applicable.

**Ethics approval** This study involves human participants and was approved by HRA and Health and Care Research Wales (HCRW) (Research Ethics Committee reference 19/IE08/0036). Participants gave informed consent to participate in the study before taking part.

**Provenance and peer review** Not commissioned; externally peer reviewed.

**Data availability statement** Data are available upon reasonable request.

**ORCID iD**
Ka Ying Bonnie Ng http://orcid.org/0000-0002-2770-6148

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
