## [Reviewer comments · BMJ Open]

ARTICLE DETAILS

TITLE (PROVISIONAL)	A discrete choice experiment exploring women's preferences in a novel device designed to monitor the womb environment and improve our understanding of reproductive disorders
AUTHORS	Ng, Ka Ying Bonnie; Evans, Rhiannon; Mentzakis, Emmanouil; Cheong, Ying

VERSION 1 – REVIEW

REVIEWER	Brooke Bullington Weill Cornell Medical College, Center for Global Health
REVIEW RETURNED	22-Nov-2021

GENERAL COMMENTS	This is an interesting and important paper, applying the DCE methodology to understand user preference for a novel intrauterine device. The paper could use some re-organization and clarification. Specifically, I think the authors could add additional rationale for the device and why it's important in the introduction. They did a good job of explaining this in the discussion, but it would help if this came sooner. Additionally, I think the authors could spend more time explaining the different choice options. For example, the "information guides treatment" attribute was confusing and needs further explanation. Further detailed comments are below. Abstract -Include more details about what novel device is in the "Objectives" section of the abstract.-The first sentence in the results section of the abstract is hard to follow. Perhaps include further clarification on what "obtaining information to guide treatment" means. Introduction -In the first paragraph of the introduction, perhaps include one further sentence about why engagement of users in development of these devices is important. Beyond just helping inform scientists, researchers, etc, it also seems that engagement of potential users would inform and benefit the users themselves.-Adding further context and explanation regarding the device and its significance would benefit readers. Table 1 -In the description for "Information obtained from the device..." include a more thorough explanation of the different levels and what they mean. I presume "all cases" would indicate that the device would be able to gather information for treatment options for anyone who uses it, regardless of their condition, but it's a little unclear.-For the levels of "Discreteness," completely discrete appears to be listed twice.
--

	Methods  -Clarify what is meant by “limited ability to understand the survey” beyond potential language barriers. -Please add a sentence justifying inclusion of all reproductive-aged women, rather than just reproductive-aged women who may have a specific disorder for which they may desire such a device. -Were participants handed a packet of papers with the explanations of the DCE and the DCE itself? Or were they explained anything orally? Please clarify in the Data collection section of the results. -Did the study team assess if women given the DCE would be interested in using the device overall? It appears this would be important for understanding the study population and could potentially affect study results. -The patient public involvement section seems repetitive, though I appreciate the inclusion of this section. Consider removing information from this section from earlier parts of the methods. Results  -Elaborate on what “had a child (either partner)” means. -It appears the study team collected information on race/ethnicity. Consider adding this to the demographics table and discussing it in the results/discussion section. -Table 4 is hard to read. Consider adding column headers or other features to make it more clear. Discussion  -In the first paragraph, it would be helpful if you summarized the findings of the study. -The third paragraph of the discussion is very strong and provides great rationale for why this device is important. Consider adding some of this rationale to the introduction of the paper, so readers have a more full understanding of the usefulness of the device.
--	--

VERSION 1 – AUTHOR RESPONSE

Reviewer 1	
Reviewer: 1 Comments to the Author: This is an interesting and important paper, applying the DCE methodology to understand user preference for a novel intrauterine device. The paper could use some re-organization and clarification. Specifically, I think the authors could add additional rationale for the device and why it's important in the introduction. They did a good job of explaining this in the discussion, but it would help if this came sooner. Additionally, I think the authors could spend more time explaining the different choice options. For example, the "information guides treatment" attribute was confusing and needs further explanation. Further detailed comments are below.	We thank the reviewer for taking the time to review our manuscript. We have addressed each comment below and have tracked the changes in the resubmission manuscript.
Abstract	We have included further details on the novel

-Include more details about what novel device is in the "Objectives" section of the abstract.	device within the objectives section of the abstract.
-The first sentence in the results section of the abstract is hard to follow. Perhaps include further clarification on what "obtaining information to guide treatment" means.	We have included more information on the levels for the attributes within the design section of the abstract.
Introduction -In the first paragraph of the introduction, perhaps include one further sentence about why engagement of users in development of these devices is important. Beyond just helping inform scientists, researchers, etc, it also seems that engagement of potential users would inform and benefit the users themselves.	We thank the reviewer for the suggestion. We have now included a sentence at the end of the first paragraph to detail this point.
-Adding further context and explanation regarding the device and its significance would benefit readers.	We have now included in the introduction further context and explanation regarding the device and its significance in advancing the care for women with reproductive disorders (paragraph 4 of introduction).
Table 1 -In the description for "Information obtained from the device..," include a more thorough explanation of the different levels and what they mean. I presume "all cases" would indicate that the device would be able to gather information for treatment options for anyone who uses it, regardless of their condition, but it's a little unclear.	Within table 1, we have now included a more thorough explanation of the different levels for the 'information obtained from the device and its use in guiding treatment' attribute.
-For the levels of "Discreteness," completely discrete appears to be listed twice.	We have now amended table 1 so that the different levels of 'discreteness' are 'completely discrete', 'moderately discrete' and 'indiscrete'. We have included the explanations to the corresponding levels.
Methods -Clarify what is meant by "limited ability to understand the survey" beyond potential language barriers.	We have now clarified this sentence and added in examples of 'limited ability to understand the survey'
-Please add a sentence justifying inclusion of all reproductive-aged women, rather than just reproductive-aged women who may have a specific disorder for which they may desire such a device.	We have included all women of reproductive age as it is beneficial to assess the preferences of all women. For future clinical studies, we will also be inserting the device into women without reproductive disorders to understand the differences in uterine environment in physiological conditions as well as pathological conditions. We have now included a sentence to justify this in the 'recruitment of participants'

	section of the methods.
-Were participants handed a packet of papers with the explanations of the DCE and the DCE itself? Or were they explained anything orally? Please clarify in the Data collection section of the results.	The participants were handed the questionnaire packs and the explanation on how to complete the questionnaire was available within the pack. Participants completed the questionnaire themselves but had oral clarification where required. This is detailed within the 'data collection' section of the methods.
-Did the study team assess if women given the DCE would be interested in using the device overall? It appears this would be important for understanding the study population and could potentially affect study results.	Within the questionnaire, we did not ask whether if the women would be interested in using the device. However, within the questionnaire pack, there was an optional feedback section where participants could write free text comments on how they feel about the device. From this, we determined that a significant proportion of study participants would be interested in using the device if it was to guide treatment in the future.
-The patient public involvement section seems repetitive, though I appreciate the inclusion of this section. Consider removing information from this section from earlier parts of the methods.	We have now shortened the PPI section within 'development of attributes and levels' and included the details within the 'patient public involvement' section.
Results -Elaborate on what "had a child (either partner)" means.	We have now clarified this within table 3 of the results and within the 'study participants' section. We have changed 'has a child (either partner)' to 'either the female participant or her partner have a child'
-It appears the study team collected information on race/ethnicity. Consider adding this to the demographics table and discussing it in the results/discussion section.	We have included a breakdown for ethnicity in the study participants table (Table 3). As over 90% of participants fall within one ethnicity category with other ethnic groups making up less than 5% each, it is not possible to make inferences across ethnicities due to low power. However, including ethnicity as a dummy variable in the latent class model (where Ethnicity =1 (British) vs Ethnicity >1 (Not British)) suggests that ethnicity is not significant in determining preference or preference heterogeneity.
-Table 4 is hard to read. Consider adding column headers or other features to make it more clear.	We have now amended Table 4 to make it clearer to the readers.
Discussion -In the first paragraph, it would be helpful if you summarized the findings of the study.	We have now summarised the findings of the study within the first paragraph of the discussion.
-The third paragraph of the discussion is very strong and provides great rationale for why this device is important. Consider adding some of this	We have provided a clinical rationale for why this device is important within the introduction section of the manuscript. We hope that this provides

rationale to the introduction of the paper, so readers have a more full understanding of the usefulness of the device.	readers with more understanding into the usefulness of the device much earlier on in the paper.
---	--